

# Topic2features: a novel framework to classify noisy and sparse textual data using LDA topic distributions

Junaid Abdul Wahid[1], Lei Shi[2], Yufei Gao[2], Bei Yang[1], Yongcai Tao[1], Lin Wei[2] and Shabir Hussain[1]

[1] School of Information Engineering, Zhengzhou University, Zhengzhou, Henan, China
[2] School of Software, Zhengzhou University, Zhengzhou, Henan, China

## ABSTRACT

In supervised machine learning, specifically in classification tasks, selecting and analyzing the feature vector to achieve better results is one of the most important tasks. Traditional methods such as comparing the features' cosine similarity and exploring the datasets manually to check which feature vector is suitable is relatively time consuming. Many classification tasks failed to achieve better classification results because of poor feature vector selection and sparseness of data. In this paper, we proposed a novel framework, topic2features (T2F), to deal with short and sparse data using the topic distributions of hidden topics gathered from dataset and converting into feature vectors to build supervised classifier. For this we leveraged the unsupervised topic modelling LDA (latent dirichlet allocation) approach to retrieve the topic distributions employed in supervised learning algorithms. We made use of labelled data and topic distributions of hidden topics that were generated from that data. We explored how the representation based on topics affect the classification performance by applying supervised classification algorithms. Additionally, we did careful evaluation on two types of datasets and compared them with baseline approaches without topic distributions and other comparable methods. The results show that our framework performs significantly better in terms of classification performance compared to the baseline(without T2F) approaches and also yields improvement in terms of F1 score compared to other compared approaches.

# INTRODUCTION

Learning to classify short text, social media data, and large web collections has been extensively studied in the past decade. Many text classification methods with a different set of features have been developed to improve the performance of classifiers and achieved satisfactory results (*Škrlj et al., 2021*). With the rapid growth of online businesses, communication, and publishing applications, textual data is available in a variety of forms, such as customer reviews, movie reviews, chats, and news feeds, etc. Dissimilar from normal documents, these type of texts have noisy data, much shorter, and consists of few sentences, therefore it poses a lot of challenges in classifying and clustering. Text classification methods

Corresponding authors
Lei Shi, shilei@zzu.edu.cn
Yufei Gao, yfgao@zzu.edu.cn

typically fail to achieve desirable performance due to sparseness in the data. Generally, text classification is a task to classify the document into one or more categories based on content and some features (*Dilawar et al., 2018*). Given a set of documents, a classifier is expected to learn a pattern of words that are appeared in the documents to classify the document into different categories. Many deep learning techniques achieve the state of art results and have become a norm in text classification tasks (*Devlin et al., 2018*), showing good results on a variety of tasks including the classification of social media data (*Tomašev et al., 2015*) and news data categorization (*Kusner et al., 2015*). Despite achieving satisfactory results on various classification tasks, deep learning is not yet optimized for different contexts such as where the number of documents in the training data is low, or document contains very short and noisy text (*Rangel et al., 2016*). To classify the data, we need a different set of features along with the data so that better classification performance can be achieved. For the classification to be successful, enough data with different features must be available to train a successful classifier (*Pavlinek & Podgorelec, 2017*). Large datasets with multiple features and labeled data do not just assure better generalization of an algorithm, but also provide satisfactory performance. However, in reality, we do not have a large number of features along with the content, and sometimes we also have few labeled instances. This norm is typical in many fields such as speech recognition, classical text mining, social media data classification (*Fiok et al., 2021*). Of course, we can do feature engineering and labeling manually but labeling is considered to be difficult and time-consuming and selection of features is unavailable when you do not have a lot of features associated with datasets (*Meng et al., 2020*). Many semi-supervised learning methods of text classification are based on less labeled data and important feature selection and focus on similarities between dependent and independent variables. Since many methods are based on analyzing the similarity measures of a label and unlabeled data, the representation of content and its features is important (*Pavlinek & Podgorelec, 2017*). The representation of unstructured content and features is more important than choosing the right machine learning algorithm (*Kurnia, Tangkuman & Girsang, 2020*). While you can represent the structured content uniformly with feature vectors, unstructured content can represent in various ways. In the text classification, some researchers leveraged vector space models representation, where features are based on words as independent units and values extracted from different vector weighting schemes such as term frequency, inverse document frequency (TF-IDF) (*Pavlinek & Podgorelec, 2017*). But in these representations, word orders and semantic meanings are ignored (*Sriurai, 2011*) that ultimately impact the classifier performance. In addition, these word vectors are sparse and high dimensional, so it is impossible to use just any machine learning algorithm on them seamlessly (*Andoni, Indyk & Razenshteyn, 2018*). For features vector representations, different techniques, such as the most common ones are TF-IDF, a bag of words, and word embeddings are utilized to fine-tune their classifiers, but sparseness remains in the representation. In this situation, we can use topic models. When we have a low number of features, topic models consider context and compact the representation of content (*Colace et al., 2014*). In this way, we can represent each document in latent topic distribution space instead of word space or document space. So inspired by the idea of, contexts in which we do not have many features,

in which we have sparseness and noisiness in data, and also the semi-supervised approaches in which we have less labeled data, we present a novel framework for text classification of various datasets of relatively same nature with hidden topics distributions retrieved from those datasets that can deal successfully with large, short, sparse and noisy social media and customer reviews datasets. The underlying approach is that we have collected datasets of different natures and then trained a classification algorithm based on a labeled training dataset and discovered topics retrieved from those datasets. The framework is mainly based on combining the unsupervised LDA topic modeling approach and powerful machine learning text classification classifiers such as MaxEnt (MaxEntropy) and SVM (Support Vector machine). This research has the following contributions:

1. We propose a novel T2F model that leverages LDA topics distributions to represent features instead of using traditional features to build classifiers. The proposed model represents features in a way to captures the context of data.
2. We have reached promising results and give a new way to solve the feature selection problem to achieve the best classification results.
3. Every aspect of model variation with different parameters analyzed in results and discussion section.

## RELATED WORK

Different studies applied various feature engineering techniques to improve the performance of classifiers. *Masood & Abbasi (2021)* used graph embeddings to classify the Twitter users into different categories of rebel users, *Go, Bhayani & Huang (2009)* used emoticons along with pos, unigram and bigram features to classify the tweet sentiments and *Kralj Novak et al. (2015)* computed the emojis sentiments, while when you see the famous topic modeling technique people have leveraged this for a variety of tasks such as event detection during disasters (*Sokolova et al., 2016*). To extract high-quality topics from short and sparse text, researchers proposed VAETM (Variational autoencoder topic model) approach (*Zhao et al., 2021*) by combining the word vectors and entity vectors representations. *Yun & Geum (2020)* used LDA-based text feature representation as an input to support vector machine classifier to classify the patent. Most of the time, topic modeling is mostly used to extract topics and analyze those topics to aid the organization in decision making (*Mutanga & Abayomi, 2020*). A recent study by *Liu, Lee & Lee (2020)* explored the topic embeddings generated from LDA to classify the email data, specifically they improved the email text classification with LDA topic modeling by converting email text into topic features. In the medical domain, *Spina et al. (2021)* proposed a method that extracts nigh time features from multisensory data by using LDA and classify COPD (chronic obstructive and pulmonary disease) disease patients, they regard LDA topic distributions as powerful predictors in classifying the data. In another approach, *Li & Suzuki (2021)* used LDA-based topic modeling document representation to fine-grained the word sense disambiguation, they proposed a Bag of sense model in which a document is a multiset of word senses and LDA topics word distributions mapped into senses. Using the text summarization techniques to label the topics generated from LDA topic distributions

is also one of the attempts made by researchers (*Cano Basave, He & Xu, 2014*). Recent work has applied summarization methods to generate topic labels. *Wan & Wang (2016)* proposed a novel method for topic labeling that runs summarization algorithms over documents relating to a topic. Four summarization algorithms are tested: TopicLexRank, MEAD, Submodular, and Summary label. Some various vector-based methods have been also applied to label the topics. *Alokaili, Aletras & Stevenson (2020)* developed a tool to measure the semantic distance between a phrase and a topic model. They proposed a sequence-to-sequence neural-based approach to name topics using distant supervision. It represents phrase labels as word distributions and approaches the label problem as an optimization problem. Recent studies have shown that similarity measures of features are more efficient when based on topic models techniques than they are based on bag of words and TF-IDF (*Xie & Xing, 2013*). In this context, the semantic similarity between two documents was also investigated (*Niraula et al., 2013*). The most related work to our context is probably the use of topic modeling features to improve the word sense disambiguation by *Li & Suzuki (2021)* and also the work in *Pavlinek & Podgorelec (2017)* in which they present features representation with a semi-supervised approach using self-training learning. As our ultimate motivation is to classify the text with good performance, so for the classification of text a lot of methods and frameworks have been developed. If we look at the aspect of feature engineering techniques, then there are a lot of mechanisms used in different studies to tune the feature for better text classification. In this way, *Nam, Lee & Shin (2015)* used the social media hashtags for sentiment classification of texts; they collected the data with the hashtags and make use of hashtags to classify the sentiments in positive and negative categories. Before the topic modeling techniques, graph embeddings were also used with n-gram features to better classify the text; *Rousseau, Kiagias & Vazirgiannis (2015)* analyzed the text categorization problem as a graph classification problem, and they represent the textual documents as a graph of words. They used a combination of n-grams and graph word representation to increase the performance of text classifiers. *Luo (2021)* leveraged the word frequency, question marks, full stops, initial word, and final word of the document. While the use of word taxonomies as means for constructing new semantic features that may improve the performance of the learned classifiers was explored by *Škrlj et al. (2021)*. In-text mining, *Elhadad, Badran & Salama (2018)* present an ontology-based web document classifier, while *Kim et al. (2018)* propose a clustering-based algorithm for document classification that also benefits from knowledge stored in the underlying ontologies.

## Topic modelling

Latent Dirichlet Allocation (LDA) first introduced by *Blei, Ng & Jordan (2003)*, is a probabilistic generative model that can be used to estimate the multinomial observations by an unsupervised learning approach. To model the topics, it is a method to perform latent semantic analysis (LSA). The main idea behind LSA is to extract the latent structure of topics or concepts from the given documents. The term latent semantic was coined by *Kim, Park & Lee (2020)* who showed that the co-occurrence of words in the documents can be used to show the semantic structure of the document and ultimately find the concept or

**Table 1  Latent Dirichlet Allocation (LDA) generalization process model.**

**For each document sample m∈M topic proportions θm from the alpha dirichlet distribution, Then for each word placeholder n in the document m, we:**

1. We randomly choose a topic Z^m,n in accordance with proportions of sample topic
2. We randomly choose a word W^m,n from the set of multinomial distributions ϕk of already chosen topic.

In the generalization process of LDA, the $\alpha$ and $\beta$ are hyper vector parameters that determine the dirichlet prior on $\theta$m is a collection of topic distributions for all the documents and on parameter $\phi$, they determine the word distributions per topic (*Pavlinek & Podgorelec, 2017*).

*Parameters and variables:*

M: total no. of documents

N: total no. of words

K: number of topics

$\phi$k: word distributions of topic K

Z^m,n: a document topic over words

W^m,n: topic words of specific document

$\alpha$: hyper vector parameter

$\beta$: hyper vector parameter

$\theta$m: topic distribution of document

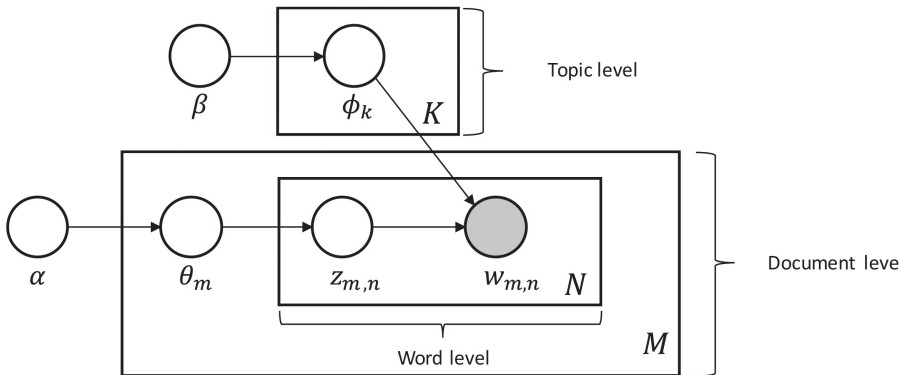

**Figure 1  Generalization of the LDA topic modelling model.**

topic. With LDA each document is represented as a multinomial distribution of topics where the topic can be seen as high-level concepts to documents. The assumption on which it is based is that document is a collection created from topics, where each topic is presented with a mixture of words. Each variable and parameter of the LDA model is defined in the Table 1.

From the above model depicted in Fig. 1, the generalization of LDA is described as follows:

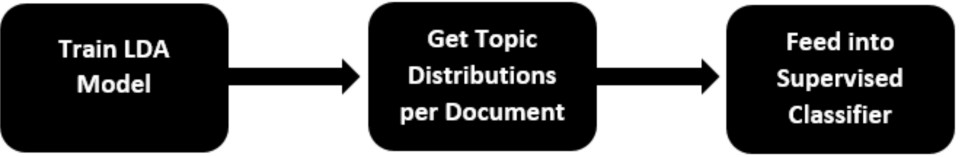

**Figure 2** Abstract model explanation of proposed framework.

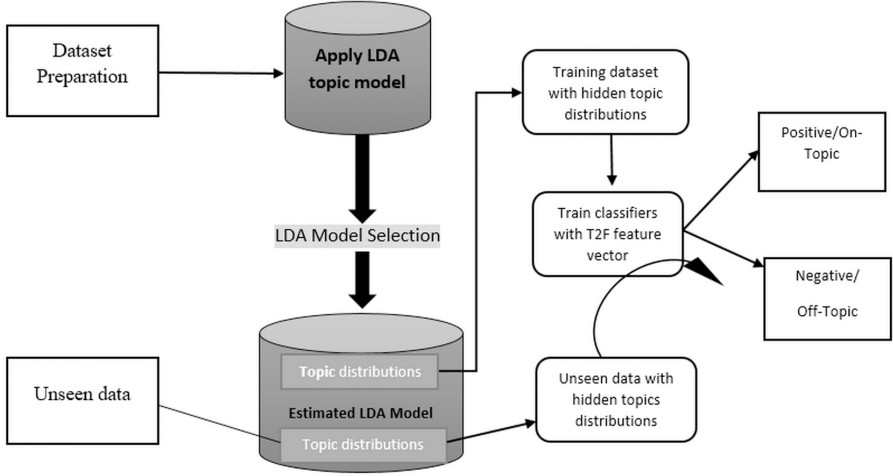

**Figure 3** Proposed framework in detail showing each steps involved.

# PROPOSED FRAMEWORK

Figure 2 shows the abstract model, which depicts the generic framework, and the detailed framework in the Fig. 3 depicted that we aim to build and train text classifiers with the use of hidden semantic topics.

The framework consist of the following tasks:

1. Collect the appropriate dataset from any domain, we choose Amazon product reviews dataset and social media dataset of different disasters.
2. Apply the LDA topic modeling with different parameters on a dataset and generate the hidden semantic topics with weights and select the appropriate LDA model.
3. Create the topic distributions for every review/tweet/document using the LDA model and convert them into feature vectors to feed in supervised algorithms.
4. Build supervised learning classifier and get F1, Accuracy, Precision, and Recall score to check the classification performance of the proposed model.
5. Also did experiment on unseen data, by applying the LDA topic distributions of current data and investigate to see if it generalizes.

The first step is more important choosing the appropriate dataset, the dataset must be large enough and rich enough to cover a variety of topics that are suited to classification problems. This means that the nature of data should be discriminative enough to be

observed by humans. The second step explains that we apply the famous topic modeling approach such as LDA (latent Dirichlet allocation) for creating topics from datasets. There is a lot of topic modeling approaches for topic modeling such as pLSA or LDA. We choose LDA because it has more concrete document generation. LDA was briefly discussed in the LDA topic modeling section. 3) As topic modeling gives the number of topics per document, we developed different LDA models with different settings such as with 10, 15, 20 topics also with the lemmatized data and using bigrams and trigrams, and also with different iterations. We observed the topic distributions outputs were impressive and satisfy our supervised learning classifiers, then we grab the topic distributions. 4) We build the classifiers by using the topic distributions as feature vectors, we choose supervised learning classifiers such as MaxEntropy, Max entropy with stochastic gradient descent (sgd) optimizer, and mostly used support vector machine (SVM). 5) This is an additional step to test the framework on unseen data, we run the classifiers on unseen data by creating topic distributions from the current LDA model and see if it generalizes or not. The extensive detail of each step will be discussed further in the relevant section.

## Datasets

Selecting the appropriate dataset is more important because the topics generated from these datasets directly impact the classifier results and performance of classifiers. Therefore to make these things in mind, we choose two large datasets of various nature. One dataset is Amazon review datasets about products and people's sentiments about the products. The total reviews were data span a period of more than 10 years, including all 500,000 reviews up to October 2012 *McAuley & Leskovec (2013)*. Another dataset was a collection of various disaster-related social media datasets (collected from *Olteanu et al. 2014*, *Imran et al., 2013*) that contains tweets from various disasters annotated based on relatedness. The tweets were collected during seven crisis occurred during 2012 and 2013 and human-made crisis or natural disaster occurred in 2016. The total tweets were 70k, with categories of different relatedness such as relevant, irrelevant, on-topic, or off-topic. Full detail of datasets given in Table 2. To check the effectiveness in various domains of our LDA models and classifiers we choose these datasets of different nature, tweet datasets are mostly short texts and noisy, and Amazon review dataset is more large text and compact detail of about products in the form of reviews.

## Data pre-processing

All the pre-processing steps are shown in Fig. 4, like removing punctuations, transforming to lowercase letters, and make into lists, the detail of remaining steps is in following section.

### *Tokenization and lemmatization*

Tokenization is the process of breaking the document or tweets into words called tokens. A token is an individual part of a sentence having some semantic values. Like Sentence hurricane is coming would be tokenized into 'hurricane', 'is', 'coming'. We have utilized the Spacy function with the core English language model for tokenization and lemmatization (https://spacy.io/usage/models). The beauty of this Spacy function is that it gives you part

| Table 2 | Dataset Statistics in detail. |
|---------|-------------------------------|
| **Dataset** | **Description** |
| Amazon user Reviews | Total 568,454 Reviews |
| | 1. 256,059 users |
| | 2. 74,528 products |
| | 3. 260 users with > 50 reviews |
| | 4. Target Categories: Positive, Negative |
| | 5. Dataset includes, Summary, Text, Sentiment score, Product ID |
| Social media Dataset | Total 70k tweets with different categories of relatedness |
| | 1. Total 7 crisis related datasets each contains 10k tweets. |
| | 2. On topic (related to crisis), off topic (not related to crisis) |
| | 3. Tweets include tweet id, tweet content, time, tweet relatedness. |

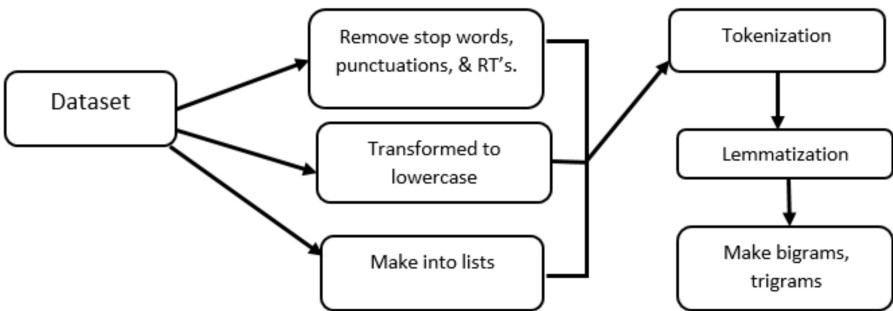

**Figure 4** Pre-processing steps involved in data pre-processing.

of speech detail of every sentence, and you can choose from that which part of speech you need for further processing in the specific context. Spacy is capable enough to also give sentence dependencies in case you need them while performing graph embeddings. After tokenization, we need to see which part of the sentence we need and also need to extract the words into their original forms. Both the lemmatization process and the stemming process are used for this purpose. Many typical text classification techniques use stemming with the help of port stemmer, and snowball stemmer, such as words 'compute', 'computer', 'computing', 'computed' would be reduced into word 'comput', a little drawback with stemming is that it reduces the word into its root form without looking into it the word is found in the dictionary of that specific language or not, as you can see 'comput' is not a dictionary word. There comes the lemmatization, Spacy; we performed the lemmatization. Lemmatization also reduced the word into its root form but by keeping in mind the dictionary database. With lemmatization the above examples of words ('compute',' computer',' computed',' computing') would be reduced to root form as ('compute',' computer',' computed',' computing'), respectively, by keeping in mind the dictionary. While implementing the lemmatization part, we keep the sentence with words

having only the 'nouns', 'adjectives', and 'verbs', which is useful if you need to be more specified about the LDA topics and in this way your topic distributions make more sense.

### Bigrams and trigrams

Sometimes in large and sparse texts, we see the nouns or adjectives that make of multiple words, so to make the semantic context of words into sentences we need bigrams and even trigrams so that they will not break into single separate unigram tokens and lost their meaning and semantic of a sentence. Bigrams is an approach to make words that are of two tokens to remain in their semantic shape so that sentence contextual meaning would not be lost. We achieved this through Genism's phrases class (https://radimrehurek.com/gensim/models/ldamodel.html), which allows us to group semantically related phrases into one token for LDA implementation. Such as ice_cream, new_york listed as single tokens. The output of genism's phrases bigram mod class is a list of lists where each one represents reviews, documents, or tweets, and strings in each list is a mix of unigrams and bigrams. In the same way, for the sake of uniformity of three phrases words tokens, we applied trigrams through genism's phrases class to group semantically related phrases into single tokens for LDA implementation. This normally mostly applies to country names such as united_states_of_America, or people's_republic_of_china, etc. The output of genism's phrases trigrams is a list of a list where each list represents review, document, or tweets, and strings in the list is a mix of, unigrams, bigrams, and trigrams. To make the LDA model more comprehensive and specific we applied the bigrams and trigrams.

### Sparse vector for LDA model

Once you have the list of lists of different bigrams, and trigrams then you pass into Genism's dictionary class. This will give the representation in the form of a word frequency count of each word in strings in the list. Genism's LDA implementation needs text as a sparse vector for the LDA model. We have used Genism's library doc2bow simply counts the occurrences of each word in documents and creates and returns the spare vectors of our text reviews to feed into the LDA model. The sparse vector [(0, 1), (1, 1)] therefore reads: in the document Human–computer interaction, the words computer (id 0) and human (id 1) appear once; the other ten dictionary words appear (implicitly) zero times.

## LDA Model
### Apply LDA Model

To apply the LDA model, there is a specific representation of the content that we need in the form of corpus and along with the corpus, we need the dictionary that assists that corpus. For different LDA models, we create a different type of corpus, with unigrams, bigrams, and trigrams. LDA model is specifically described in detail in Fig. 1.

### LDA Model selection

LDA model selection was the most difficult task, as it can ultimately impact the results of supervised classifiers. Therefore, to choose the best LDA model with many numbers of topics in the model was a time-consuming task. Finding the exact number of topics

suited for a better LDA model was the main focus of previous studies (*Greene, O'Callaghan & Cunningham, 2014*). The first technique was manual, which is to choose the different number of range of topics and check and investigate the results if it makes any sense. The second one was analyzing the coherence score metrics of LDA models more coherence increase means better model. Then we also explored the models by giving several various topics and every topic distributions results and vectors feed into supervised algorithms and check which one gives the better results in terms of F1 score. Approach one was very time consuming, the second one was to see the coherence score but that just check the topic identifications have not a large impact on supervised algorithms results, the third approach seems suitable in our context but our main purpose was to classify the documents/reviews with best results. Genism also provides a Hierarchical Dirichlet process (HDP) (https://radimrehurek.com/gensim/models/hdpmodel.html) class that used to seek the correct number of topics for a different type of datasets it is not necessary to type the number of topics in HDP class and automatically seeks the number of topics based on data. It is only necessary to run this for a few times, and if it provides the same results with the same number of topics again and again then those number of topics are perfect learning topics for your type of data.

### Hierarchical Dirichlet process

According to Genism's documents, the hierarchical Dirichlet process (HDP) is based on stick-breaking construction that is an analogy used in the Chinese restaurant process. For example, in Fig. 5, we need to assign 8 to any of the topics C1, C2, C3. There is a 3/8 probability that 8 will land in topic C1 topic, 4/8 probability that 8 will land in C2 topic, there is 1/3 probability that 8 will land in topic C3. HDP coherently discovered the topics, like bigger the cluster is the more likely it for the word to join that cluster of topics. It is a good way to choose a fixed set of topics for the LDA model (*Wang, Paisley & Blei, 2011*). While implementing HDP on our datasets, we test our third approach as well which was manually give the topic number and check the classifier results, to ensure consistency we built around 15 LDA models with different parameters, and compared with HDP results, and choose the best ones, that has a high influence on classifiers results and that gives best classifier results. In the end, the best LDA models were with lemmatized texts (with nouns, adjectives, and verbs), with 100 iterations, and with 10, 15, and 20 topics.

## Train classifiers with topic distributions

The method for training the classifiers with topic distributions contains these steps: first, we choose the text classification algorithm from different learning methods; second we incorporate the topic distributions with some manually engineered features into the training data, test data, and future unseen data with specified representation that classifier needed. Then, in the end, we train the classifier and get the F1 measure scores.

## Choose classical text classification learning methods

We have chosen the logistic regression classifier aka MaxEnt (MaxEntropy) and Support vector machines (SVM) to evaluate our framework. The reason for choosing these classifiers is that our implementation of topic distribution works with data represented as dense or

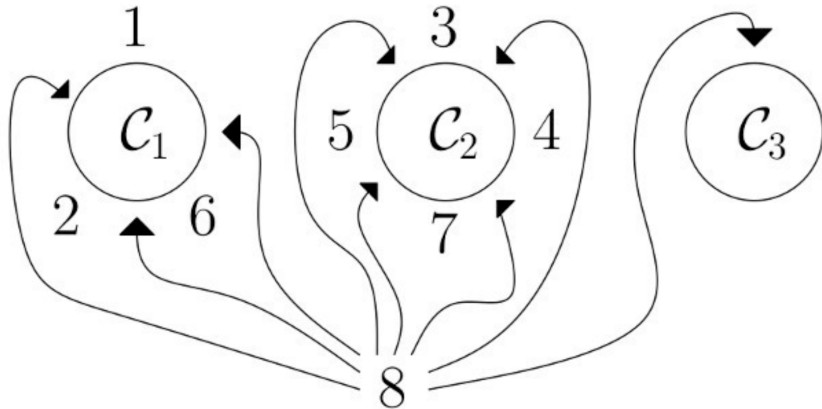

**Figure 5** Chinese restaurant analogy: HDP process based on Chinese restaurant analogy; in this analogy, C1,C2,C3 are tables and surrounding them are customers (1, 2,…,7), and new customer 8 needs to be assigned to any of the tables; so there is a 3/8 probability that the customer will be assigned to C1, a 4/8 probability 8 will be assigned to C2 and a 1/8 probability that the customer will be assigned to C3.

sparse arrays of floating-point values for the feature vectors. Therefore, these models fit with this type of implementation context and can handle the sparse and dense type of feature arrays with floating values. Also, MaxEnt makes no independent assumptions for its features like uni-grams, bigrams, and trigrams. It implies that we can add features and phrases to MaxEnt without the fear of feature overlapping (*Go, Bhayani & Huang, 2009*).

## Integrate topic distributions into dataset

After implementing the LDA model on data and getting topics from the data, we created the topic distributions and incorporate the topic distribution, original document and one manually coded feature which is the frequency of document, into the classifier in a way that resulting vector representation would be according to the machine learning classifier format. Given a dataset $W = w^{m\ k}$, for example we need to classify w from a collection of documents $W.w$ can be training, testing, or unseen data. Topic extraction for w needs to perform LDA. However, the number of iterations for inference is much smaller than of parameter estimation. The topic inference/extraction is demonstrated in the LDA generalization process in Fig. 1, here in algorithm 1, we explain how we integrate those topics into feature vectors.

This algorithm consists of two main components, first, it creates the topic distribution in the form of probability and the second one is to convert those topic probability distributions along with the length of each document to create topic-oriented feature embeddings. As presented in the algorithm we intended to learn topic-based feature embeddings to be used in classifiers. We began by creating topic distributions for all the documents in the dataset, which are nothing but word distributions along with their weights, subsequently, we convert these topics into the format to create feature vectors that are ultimately used in classifiers. For this, it utilized the get document topic function to be applied on extracted topic words (line 7), which gives output in the form of integer and float values of each

topic, after the algorithm learns the topic distributions float values and mapped into feature vectors embeddings to be used in the classifier (lines nine to 11).

---

**Algorithm 1:** Topic2vec: Integrating topic distributions into feature vectors

---

**Input**: Dataset in form of document and tweets

**Output**: topic embeddings feature vector to be used in classifier

1  $ki$= Topic for each document;

2  Initialization;

3  topics ($ki$) ← LDA (pre-processed text (tweets/documents));

4  create topic distributions ($ki$) ← Dataset;

5  **while** *not the end of document* **do**

6   create topic embeddings;

7   **for** *each document* **do**

8    topic distributions = get document topic(topics);

9   **end**

10   **for** *each topic distributions* **do**

11    topic distributions ← feature vector(topic dist length of documents);

12    feature vectors ← array(topic dist and length of document);

13   **end**

14  **end**

---

After we doing topic inference through LDA, we will integrate the topic distributions $tdm = tdm,1….., tdm,2….., tdm, k$ and original document $di = dim,1….dim,2….., dim, n$ in a order that resulting vector is suitable for the chosen learning technique. Because our classifier only can take discrete feature attributes, so we need to convert our topic distributions into the form of discrete values. Here we describe how we integrate the topic distribution into documents to get the resultant feature vector to be used into classifiers. Because our classifiers require discrete feature attributes so it is necessary to discretized probability values in $tdm$ obtained topic names. The topic name appears once or several times depending on its probability. For example, a topic with probability 0.016671937 appears 6 times will be denoted as 0.016671937:6. Here is a simple example of integrating the topic distribution into its bag of words vector to obtain the resultant vector.

$tdm = tdm, 1….., tdm,2….., tdm, k$; where $tdm$ = topic distribution of a single document

$di = dim,1…. dim,2….. , dim,n$; where $di$ = documents/reviews/tweets from dataset

Where each Topic distribution ($t^{\mathrm{dm}}, k$) is computed as follows:

$$t_{dm}k = \frac{n_m^k}{\sum_{j=1}^{k} n_m^j} \qquad (1)$$

where

 $n_m^k$ =number of words in documents assigned to topic ($k$),

 and $n_m^j$ = total no. of words in document ($m$),

$di$m = [confection, century, light, pillow, citrus, gelatin, nuts, case, filbert, , chewy, flavorful, yummy, brother, sister] and $td^m$: [0.18338655 ($td1$), 0.18334754 ($td2$), …., …., …., 0.016671937 ($tdn$) ,…..] . Applying discretization intervals

$$t_{dm} \cup td^m = rv$$

where $rv$ (resultant vector to be used in classifier) = [[confection, century, light, pillow, citrus, gelatin, nuts, case, filbert, chewy, flavorful, yummy, brother, sister]], Topic1: Topic2: Topic3: Topic3: Topic4: Topic3: Topic3:Topic3: Topic8: Topic9: Topic10

We built multiple LDA with bigrams, trigrams, with different ranges of topics, and ultimately we estimated the best LDA model with the best hyper-parameters setting, and that yields better results when fed into supervised algorithms. The best one is 20 topics, with 100 iterations, and with bigrams. For this, we use the LDA get _document_topic function from Genism's library (https://radimrehurek.com/gensim/models/ldamodel.html) on our topic distributions and get the topic distributions in the form of discretized probability values. Extracted LDA topics make the data more related, these are nothing but the probability distributions of words from documents, we built multiple topic models with various settings. We are more interested in seeing how the hidden topics' semantic structure can be converted into and applied on a supervised algorithm and to see if it can improve the performance of supervised classification.

## Train classifiers

We trained support vector machines classifiers and MaxEnt with stochastic gradient descent (sgd) optimization as it gives good results in terms of speed and performance. We have investigated that the Amazon review dataset has a disproportionate amount of classes, so on the Amazon dataset to handle the class imbalance we use the parameter class weight with value balanced. This will approximate under-sampling to correct for this. Besides this, all classifiers were applied with the same parameters. One thing to note here; while we were implementing these classifiers we noticed a modified Huber loss option in stochastic gradient implementation; the benefit of using this is that it avoids misclassification and it punishes you more on outliers as it brings tolerance to outliers as well as probability estimates. Therefore, we utilized this parameter to avoid misclassifications and punishing the outliers.

## Evaluation

To verify our proposed framework we performed two classification tasks with two types of datasets, the first task was to check the sentiments of people from Amazon reviews, in short, classify the reviews into different categories of sentiments, the second task was classification of social media text into different categories of relatedness (on topic, off-topic, relevant, irrelevant) during natural crisis and disasters. Tweet texts are very short in comparison to reviews of Amazon datasets, each includes tweet id, tweet, tweet time and label. Amazon reviews are a bit longer. Each contains several sentences and also describes particular features of various products. Both datasets were sparse, short text, noisy, and hard enough to verify our framework.

## Evaluation measures

Typically classification algorithms have the accuracy, F1 measure, Precision, and recall measures to measure the performance of the model. Accuracy is a measure to identify all correctly classified categories. Precision is a measure to identify positive from all predicted positive classes, while recall is a measure to correctly identify positive classes from actual positive classes, and F1 is a harmonic mean of precision and recall (*Elhadad, Badran & Salama, 2018*), (https://medium.com/analytics-vidhya/accuracy-vs-f1-score-6258237beca2). So, to evaluate the performance of our proposed framework, we used the F1, Precision, and recall measure scores, because during preprocessing of our data we have investigated that our datasets have imbalanced classes, and F1 measure score is a suitable metric for imbalanced classes' datasets. Besides F-measure, precision, and recall scores, we also intended to measure the statistical significance of our model, so we employed a k-fold cross validation (CV) test on our proposed model and as well on baseline approaches. We determine the classification accuracy of each fold on our datasets, and evaluate the average classification accuracy of our proposed framework and compared it with baseline approaches, to check the effectiveness of our model. The major advantage of k fold CV is that it takes every observation of data to have a chance of appearing in the training and testing set. The higher the mean performance of the model, the better the model is, therefore mean accuracies on k fold CV and average F1 score are dominantly used as evaluation measures.

### Statistical Validity test

In order to have statistical validity of our model and compare it with a baseline to observe any significant difference in performance, we ran a 5x2cv paired $t$-test on our dataset. Although there are many statistical tests, we applied this because it is a paired test, and in machine learning this means that the test data for the baseline and the trained model are the same, in our context, it is the same; we used the same Amazon and social media datasets for the baseline and proposed model. As its name implies this test typically split the dataset into two parts (training and testing) and repeat the splitting(50% training and 50% testing) five times, in each iteration (*Dietterich, 1998*). In each of the five iterations, we fit A and B to the training split and evaluate their performance ($pA$ and $pB$) on the test split. After this, it again rotates the test and train sets and computes performance again, which results in 2 performance difference measures:

$$p^1 = p_A^1 - p_A^1 \qquad (2)$$

$$p^2 = p_A^2 - p_A^2 \qquad (3)$$

Then it estimates the estimate mean and variance of differences through following equations;
 mean is:

$$\bar{p} = \frac{p^1 + p^2}{2} \qquad (4)$$

and variance is:

$$s^2 = (p^1 - \overline{p}^2)^2 + (p^2 - \overline{p}^2)^2 \tag{5}$$

The formula of computing $t$-test statistics for this test is as follows:

$$t = \frac{p1^1}{\sqrt{1/5\sum_{i=1}^{5} S_i^2}} \tag{6}$$

where $p1^1$ is p1 from very first iteration. The t statistics assuming that it approximately follows as t distribution with 5 degrees of freedom, and our hypotheses statements and threshold values are;

H0 = Both the classifiers have same performance on this dataset.

H1 = Both classifiers does not have same performance on this dataset.

Our threshold significance level $\alpha = $**0.05** for rejecting the null hypothesis that both classifiers have same performance on this dataset. Under the null hypotheses, t-statistics value approximately follows a t-distribution with 5 degrees of freedom, so its value should remain in a given confidence interval which is **2.571** for **5%** threshold, and it indicates that both classifiers have equal performance, if t-statistics value greater than this value, we can reject the null hypotheses. You can implement the 5 by 2 fold cv paired $t$-test from scratch, but there is a package called MLxtend that implements this test and gives you t-values and $p$-values of two models (http://rasbt.github.io/mlxtend/user_guide/evaluate/paired_ttest_5x2cv/), in its parameters, we just gave the models names and scoring mode was mean accuracy.

## Amazon reviews sentiment classification

Sentiment analysis is a typical classification problem, used in various ways, some researchers apply sentiment analysis on reviews of movies (*Shen et al., 2020*). Many deep learning and natural language processing techniques are proposed for sentiment analysis (*Ullah et al., 2020*). For sentiment classification in our article, we have considered a public dataset that we collected from the data repository Kaggle. The dataset description is already given in the dataset section, which is the collection of customer reviews of customers about Amazon products. The reviews are assigned into two categories positive and negative. 1/5 of the total reviews we used as test data and the remaining used as training data. We utilized retrieved bigrams, trigrams, and lemmatized text from the dataset and apply the LDA model with different parameters, and with 10, 15, and 20 topics. One thing is to be noted here is that we lemmatized the data and take only nouns, adjectives, and verbs to grab the actual meaning from reviews and apply the LDA model to actual contextual meaning of texts or reviews. After the lemmatization process, it remains with 378,123 reviews.

### Result and analysis of Amazon dataset

To examine our proposed models based on evaluation measures with different parameters settings, we examined the F1, precision, and recall scores of classifiers.

We randomly divided the data into 5-fold CV, we ran experiments by feeding different LDA topic distributions into the classifiers. The results are in the Figs. 6, 7 and 8, on $Y$-axis

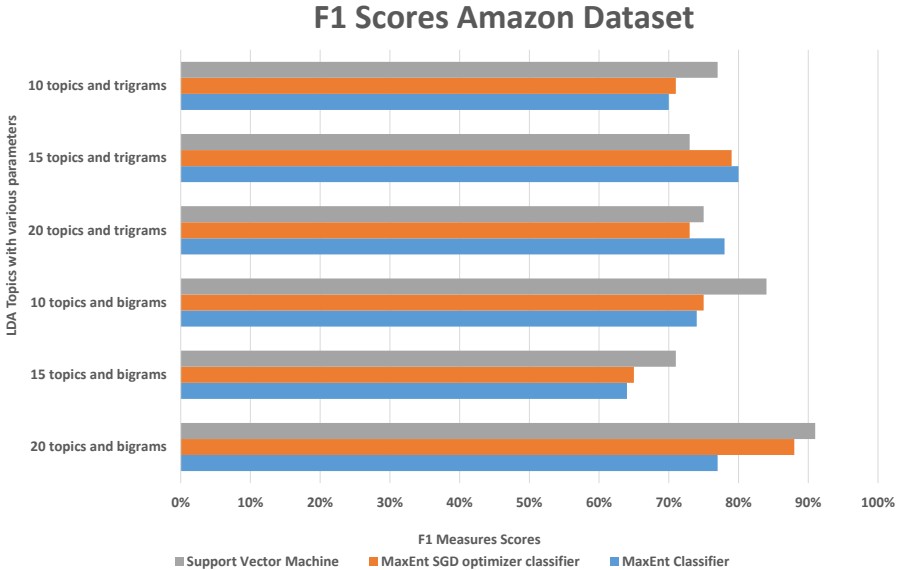

**Figure 6** Comparison of the Amazon dataset f1 measure scores with different parameters.

the LDA models with different parameters, and on *X*-axis it shows the classification results with each classification algorithms. The best performing algorithms (using bigrams and 20 topics) based on the precision scores are MaxEnt, SGD, and SVM. As seen in Fig. 6, these two algorithms (with bigrams features and 20 topic distributions) are slightly better than the other algorithm. In Fig. 7, a noise factor can be seen as MaxEnt and MaxEnt SGD underperformed in terms of recall scores. Then we can see, the model with bigrams and 20 topics achieved the highest F1, precision, and recall score of 91% with support vector machine classifier, while when we apply the trigrams into the LDA model, the MaxEnt classifier algorithm achieved the best result. It implies that the Amazon review dataset has large texts and when you lower the topics then it works well with trigrams and when you increase the topics it works well with bigrams.

### Comparison with baseline approaches Amazon dataset

Below in the Table 3 we provided the result without the LDA model, we compare the result with baseline approaches by classifying the data without leveraging the topic distributions. We have implemented the most commonly used TFIDF feature vectors with different classifiers on our Amazon review dataset as a baseline. When we apply the classifiers with TFIDF feature vector representations then F1 scores decrease about 9% and 17% with support vector machine and Multinomial Naive Bayes classifiers respectively as compared to T2F with support vector machine. This means that topic distributions give better results because it semantically capture the words within the documents and their distributions, so that classification performance would be increased, and our proposed framework able to achieve the higher classification results than the baseline approaches. While TFIDF has been popular in its regard, there remains a void where understanding the context of the word was concerned, this is where word embedding techniques such as doc2vec can be

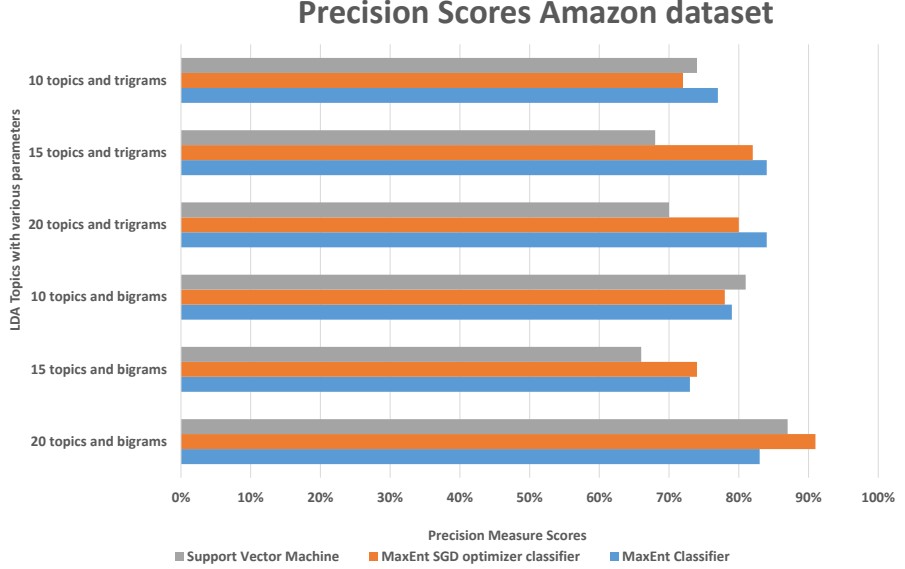

**Figure 7** Comparison of the Amazon dataset precision scores with different parameters.

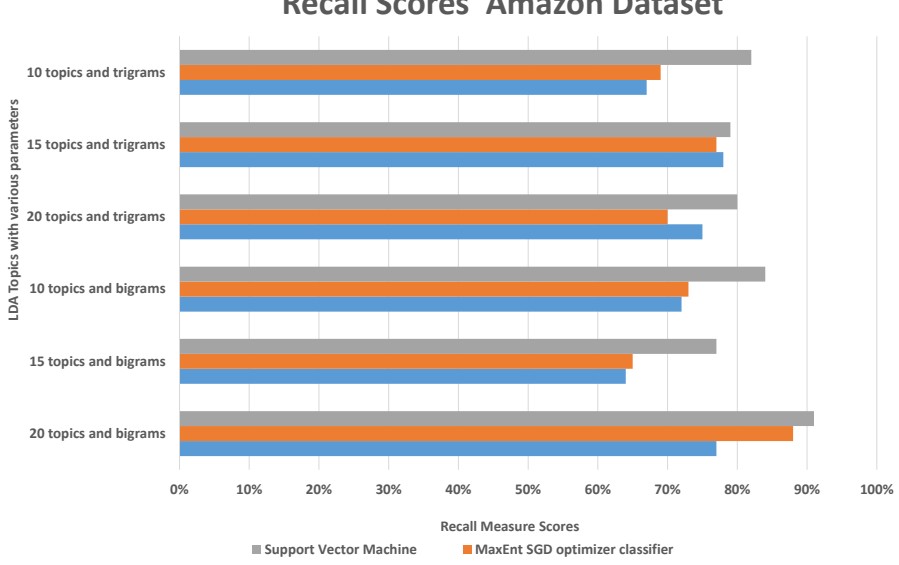

**Figure 8** Comparison of the Amazon dataset recall scores with different parameters.

utilized, *Le & Mikolov (2014)*. Therefore, we implemented doc2vec with logistic regression classifier as one of our baseline approaches to analyze if it increases performance, the f1 score reaches 86% as compared to the TFIDF approach with SVM classifier which was 82%, but still lower than the F1 score of 91% which we achieved by applying LDA topic distributions as feature vectors.

**Table 3** Amazon Dataset F1, Precision,Recall and Average accuracy statistics: comparison with baseline approaches evaluation measures results.

| Algorithms | F1 score | Precision | Recall | Mean Accuracy |
|---|---|---|---|---|
| SVM (TFIDF) | 82% | **83%** | 80% | 74% |
| Multinomial Naive Bayes(TFIDF) | 74% | 76% | 75% | 71% |
| MaxEnt (TFIDF) | 71% | 72% | 68% | 73% |
| MaxEnt (doc2vec) | **86%** | 77% | 90% | 79% |
| MaxEnt Sgd (proposed T2F) | 88% | **91%** | 88% | **81%** |
| MaxEnt (proposed T2F) | 77% | 83% | 77% | 73% |
| SVM (proposed T2F) | **91%** | 87% | **91%** | 77% |

To examine the classification accuracy and compared it with baseline approaches, we performed 5-fold CV in which we reserved 1/4 observations as the validation set and 4/5 as training observations the advantage of leveraging 5 fold CV is it uses every sample of the dataset in training and testing in iterations. We ran 5-fold CV experiments on the baseline approaches to measure the classification accuracy and also on the proposed model. The detailed accuracy is also shown in Table 3. The comparison of accuracy shown in Fig. 9, starting from fold 1 our approach performs less than the baseline, but after fold 1 it performs better on each fold than the applied baseline and overall average classification accuracy is also higher, the last two columns show the average classification accuracy which improved from 79.3% to 81%, i.e., classification error reduces from 20.7% to 19%. This means that within the dataset with a certain degree of words shared among the documents our framework is capable to reduce the classification error and increase the classification mean accuracy.

To compare the proposed model with the applied baselines approaches and check which approach has more statistical significance on the same Amazon dataset, we ran the 5 X 2 CV paired test on models, and compare the applied baselines with the proposed MaxEnt sgd model, we compared MaxEnt sgd because, if we see the Table 3, the mean accuracy of MaxEnt sgd is higher than other proposed models. We computed the 5X2 CV paired t-test's $t$-value and $p$-value of models, then compare it in the following table. We computed every fold(2 folds) of each iteration(5 iterations) and listed the mean results in the table. You can see in the Table 4, proposed MaxEnt sgd comparison with every baseline model has $p$-value less than the threshold value $\alpha = 0.05$, and also t-statistics value is greater than the threshold value, thus we can reject the null hypotheses and accept that two models have significantly different performance, and T2F with MaxEnt sgd with better mean accuracy has performed significantly better than the applied baselines.

## Social media data classification

To find out how our method works well with another kind of data and in different domains, we leveraged the social media datasets from the domain of disasters. We performed experiments with tweet classification with the categories of on the topic, and off-topic, support government, criticize the government. For the sake of simplicity, we take off-topic

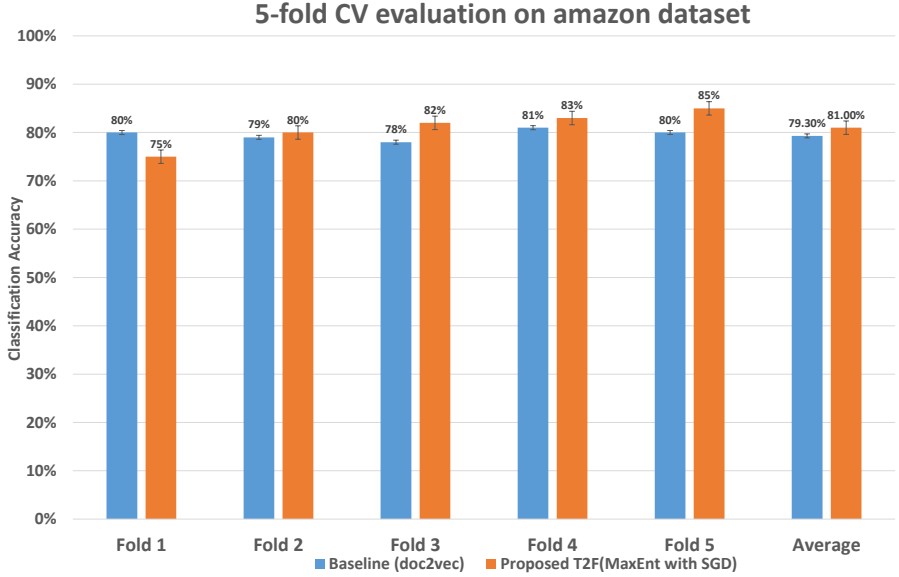

**Figure 9  Classification accuracy comparison between baseline and proposed approach on the Amazon dataset.** The 5-fold CV scores of best performing classifier of baseline and best performing classifier of proposed are shown, demonstrating each fold results of classifiers and comparing those classifiers who achieved highest classification average accuracy.

**Table 4  Comparison of each baseline model with the Proposed MaxEnt sgd(T2F) on the same Amazon dataset, and the *t*-value and *p*-value scores are listed.**

| Algorithms | MaxEnt sgd(T2F) t-statictics value | MaxEnt sgd(T2F) *p*-value |
|---|---|---|
| SVM (TFIDF) | 3.248 | 0.0437 |
| Multinomial Naive Bayes(TFIDF) | 4.784 | 0.0079 |
| MaxEnt (TFIDF) | 4.562 | 0.0060 |
| MaxEnt (doc2vec) | 2.932 | 0.0362 |

and on-topic categories. On-topic means tweet is related to and within the context of a specific disaster, similarly, off-topic means tweet text is not about the disaster. There are numerous applications of classification in the context of natural disasters or pandemics such as classify the situational information from Twitter in pandemics (*Li et al., 2020*). Some researchers utilized the topic modeling techniques and analyze the topics during disasters by leveraging Twitter data (*Karami et al., 2020*). As social media is one of the main and user-oriented text data sources therefore we have utilized the social media datasets to check the efficiency of our framework. After the pre-processing steps, we remain with 61,220 tweets.

### Result and analysis of Social media disaster dataset

As in the Amazon dataset, we ran the same LDA models on the social media datasets, and investigate the F1, precision, and recall scores, social media data is more difficult to classify, as it has more slang words, therefore we can see in Fig. 10, the false positive and

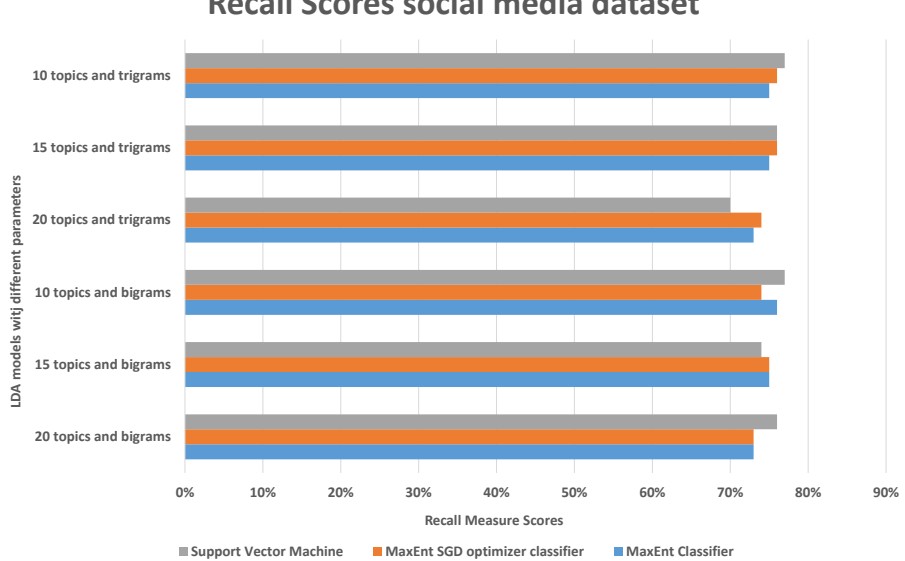

**Figure 10** Comparison of recall scores of the social media dataset with different parameters.

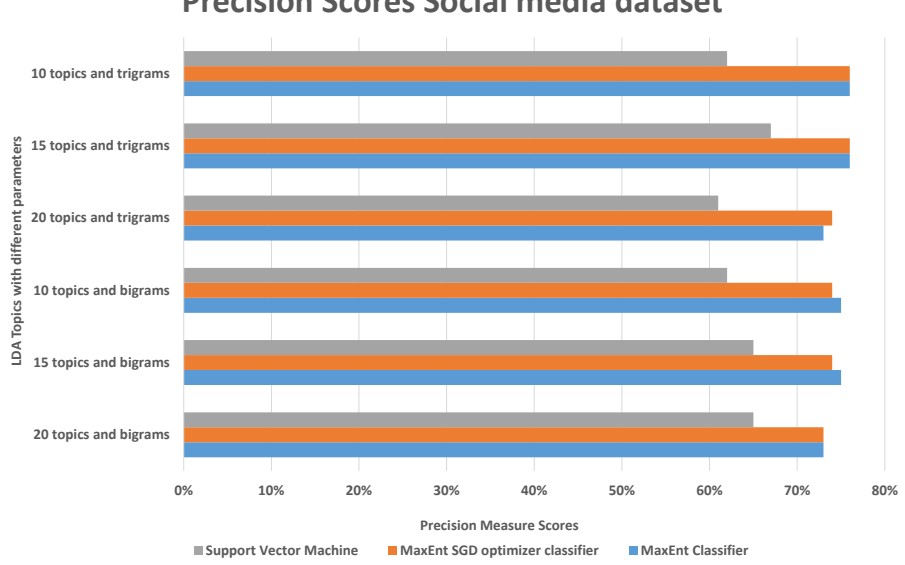

**Figure 11** Comparison of precision scores of the social media dataset with different parameters.

false negative (recall) is higher than the precision with support vector machine algorithm, in Fig. 11 in precision scores we can investigate rather than with the 20 topics it relatively gives better result with 15 topics and 10 topics with MaxEnt and MaxEnt sgd classifiers. With both bigrams and trigrams setting and 10 and 15 topics, it performs best with the MaxEnt sgd classifier giving up to 78% F1 score as shown in Fig. 12, it may be because MaxEnt sgd classifier can well handle the noisy short and sparse type of data (*Go, Bhayani*

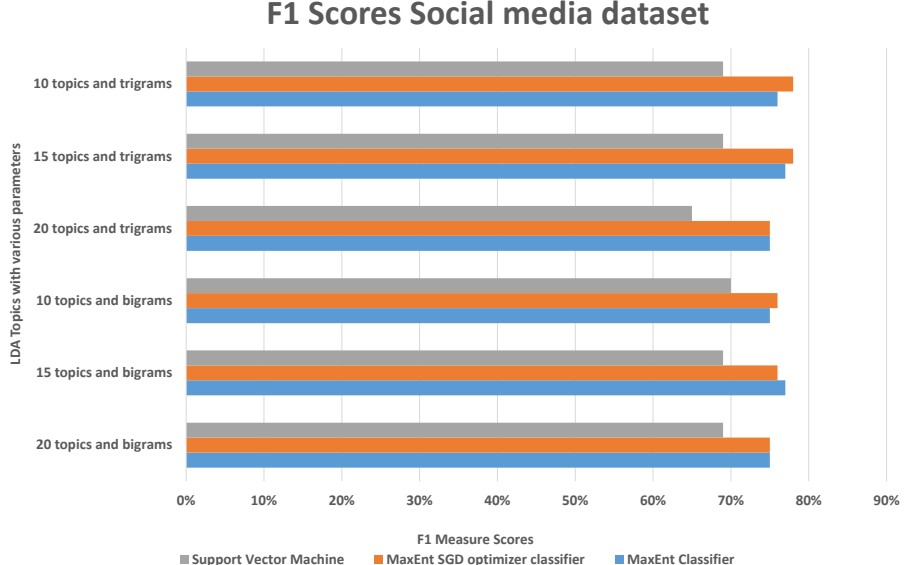

**Figure 12** Comparison of F1 scores of social media dataset with different parameters.

*& Huang, 2009*) thus having a higher coverage, and social media is the same kind of noisy unstructured text data. Also in a two-class scenario, it works well because of the binary nature of the target class and we have target class is binary in the social media dataset. Interestingly support vector machines outperformed others algorithms in terms of recall that implies that support vector machines can also somehow, if not at all, handle the noisy data of social media. The social media dataset was short and noisy such as it contains slang, etc. Therefore, it can be seen that the highest F1 score with any parameter setting reached up to 78% as compared to 91% with the Amazon dataset. But that is also satisfactory in the context of the social media data with unsupervised topic modeling. As compared to the Amazon dataset's large texts, the social media dataset gives more satisfaction with fewer topics, this is because of the length of tweets, which implies that the LDA topic model with less topic setting gives more good results than the LDA topic model with more topics.

### Comparison with baseline approaches on social media dataset

In comparison with the baseline approaches that we implemented with different types of word embedding techniques such as TFIDF and doc2vec when applying on social media dataset, it reaches up to 75% high in terms of F1 score with doc2vec embeddings on logistic regression classifier, but still less than the overall highest 78% F1 score with topic distributions as feature vectors, which shows how topic distributions accurately capture the contextual meaning and classify he data accurately. However, an interesting aspect is to analyze that F1, precision, and recall score increases while implementing doc2vec embeddings which indicates among the baseline approaches doc2vec performs best.

As we had run experiments on Amazon datasets same we run 5 fold CV on social media dataset to determine the classification significance by comparing the mean accuracy, although the classification means accuracy drops as compared to when applying on

## 5-fold CV evaluation on socia media dataset

**Figure 13** **Classification accuracy comparison between baseline and proposed approach on social media dataset.** The five fold CV scores of the best performing classifier of baseline and best performing classifier of the proposed model are shown, demonstrating each fold results of classifiers and comparing those classifiers who achieved highest classification average accuracy.

Amazon dataset, still it gives 73% mean accuracy with proposed T2F approach on MaxEnt sgd classifier, when it compared to baseline approaches it falls to 69% with doc2vec feature on MaxEnt classifier, which is highest among the baseline approaches only, but still lower than the proposed T2F approach. The mean accuracy of each fold comparison of highest baseline and highest proposed given in Fig. 13, it starts from fold 1 to fold 5 then, in the end, last two bars showing the mean accuracy which depicts how the classifiers feed with the topic distribution features classified the data significantly better than the baseline approaches and also even better than the mostly used NLP deep learning baseline approach doc2vec.

In order to compare the proposed model with the applied baselines approaches and check which approach has more statistical significance on the same social media dataset, we ran the 5X2 CV paired test on models, and compared the applied baselines with the proposed MaxEnt sgd model; we also compared MaxEnt sgd because also on the social media dataset (see Table 5), the mean accuracy of MaxEnt sgd is higher among the proposed models. We computed the 5X2 cv paired t-test's $t$-value and $p$-value of models, then compare it in the following table. We computed every fold (two folds) of each iteration (five iterations) and listed the mean results in the table. As shown in Table 6, the proposed MaxEnt sgd comparison with every baseline model has $p$-value less than the threshold value which is $\alpha = 0.05$, and also t-statistics value is greater than the threshold value, thus we can reject the null hypotheses and accept that two models have significantly different performance, and

**Table 5  Social media dataset F1, Precision, Recall and Average accuracy statistics: comparison with baseline approaches evaluation measures.**

| Algorithms | F1 score | Precision | Recall | Mean Accuracy |
|---|---|---|---|---|
| SVM (TFIDF) | 68% | 71% | 70% | 67% |
| Multinomial Naive Bayes (TFIDF) | 73% | 76% | 74% | 68% |
| MaxEnt (TFIDF) | 52% | 60% | 54% | 65% |
| MaxEnt (doc2vec) | 77% | 75% | 74% | 69% |
| MaxEnt Sgd (proposed T2F) | **78%** | 76% | 76% | **73%** |
| MaxEnt (proposed T2F) | 77% | **76%** | 76% | 69% |
| SVM (proposed T2F) | 70% | 65% | **77%** | 68% |

**Table 6  Comparison of each baseline model with the proposed MaxEnt sgd (T2F) model on same social media dataset, and the $t$-value and $p$-value scores are listed.**

| Algorithms | MaxEnt sgd(T2F) t-staticics value | MaxEnt sgd(T2F) $p$-value |
|---|---|---|
| SVM (TFIDF) | 3.257 | 0.0083 |
| Multinomial Naive Bayes (TFIDF) | 3.127 | 0.0024 |
| MaxEnt (TFIDF) | 4.273 | 0.0071 |
| MaxEnt (doc2vec) | 3.101 | 0.0271 |

proposed MaxEnt sgd(T2F) with better mean accuracy has performed significantly better than the applied baselines.

## Result and analysis of unseen data

To further investigate the efficiency of our framework we validate the LDA model on completely unseen data, for this, we chose the Amazon dataset that has data of reviews on yearly basis, we prepared the LDA model of 2011 data and use the same model to get feature topic distributions for 2012 data, it is to be noted that LDA model did not see this 2012 data, it is completely unseen for the trained LDA model. We get the test vectors for 2012 data and re-run the classifiers, results are reasonably well, as you can see in Fig. 14, it gives 87% F1 score, 87% precision score with support vector machine classifier, and 79% F1 score with MaxEnt classifier and 81% F1 score with MaxEnt sgd classifier. The 5-fold CV test was also applied to this data and in classification accuracy, the SVM classifier gives the best classification performance with 83% mean accuracy. Also each fold unseen data test results shown in Fig. 15 with all three classifiers. This also implies even if the model did not see the data, it classified it with good classification accuracy and F1 scores. Results indicate that this framework also works well with unseen data of the same context. We did the validity tests through 5-fold CV and through Mcnemar's test by using the model trained on 2011 data and test it on 2012 unseen data.

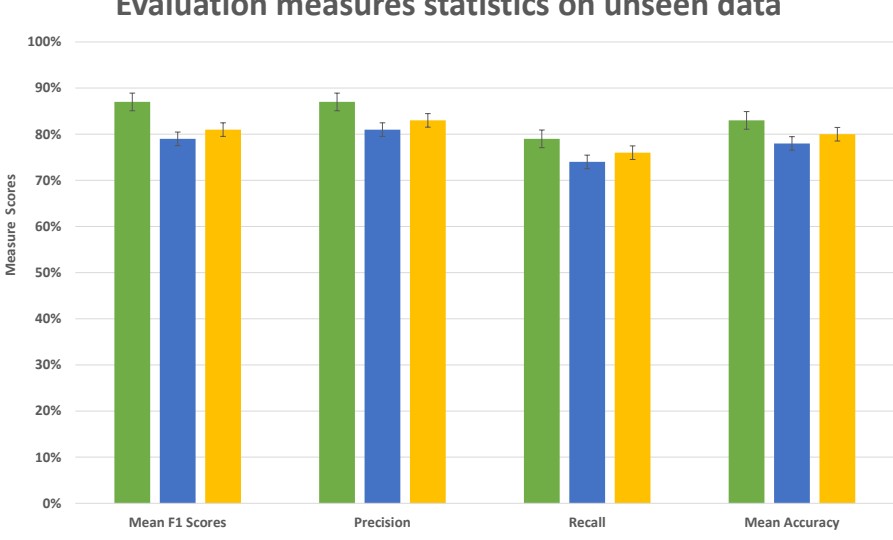

**Figure 14** Comparative results of evaluation measures statistics on unseen data.

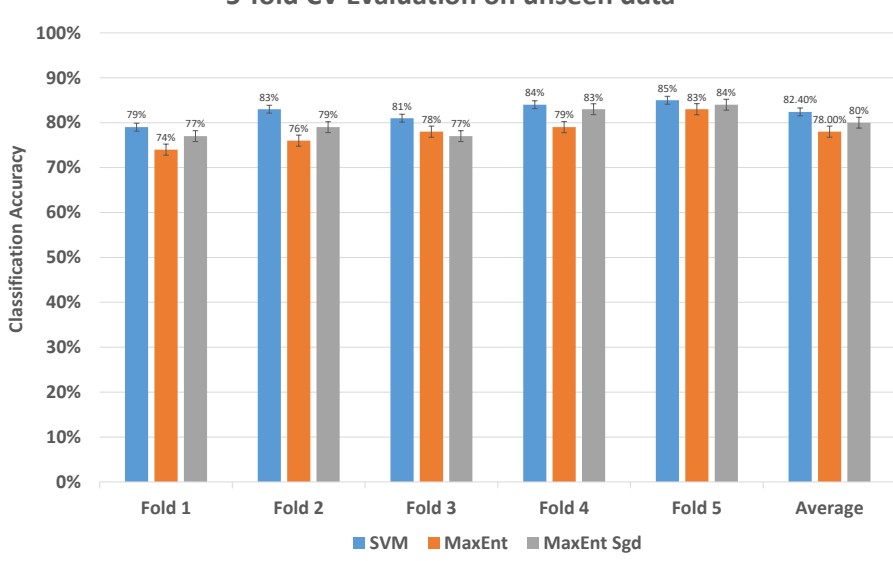

**Figure 15** Comparative results of each fold of all models on unseen data. The every fold result of our model that we applied to data unseen data; by using the train vectors of previous data, we applied to this unseen data to check the validity of our models that was trained on previous data.

### McNemar's statistical test on unseen data

We did a hypotheses test by applying McNemar's test to check whether these classifiers are statistically significant on unseen data. In machine learning McNemar's test can be used to compare the performance accuracy of two models (*McNemar, 1947*;

**Table 7  Contingency table for McNemar's statistics test.**

| correctly classified by both A and B (n00) | correctly classified by A but not by B (n01) |
|---|---|
| correctly classified by B but not by A (n10) | correctly classified neither by A or B (n11) |

http://rasbt.github.io/mlxtend/user_guide/evaluate/mcnemar/). McNemar's test operates on contingency table values that showed in Table 7.

A = SVM classifier

B = MaxEnt sgd classifier

The McNemar's test is computed as follows:

$$\frac{(|n01 - n10| - 1)^2}{n01 + n10} \tag{7}$$

n00 = no of samples correctly classified by both A and B

n01 = no of samples correctly classified by A but not by B

n10 = no of samples correctly classified by B but not by A

n11 = no of samples not correctly classified by either A or B

The first step in the statistical test to state the Null hypotheses statement. Our statement is;

H0 = cannot reject the null hypotheses indicating both classifiers have the same performance on the dataset if the calculated $p$-value greater than the threshold $p$-value.

H1 = can reject the null hypotheses indicating both classifiers have different performance on dataset if calculated $p$-value less than threshold $p$-value.

We ran chi-squared McNemar's with threshold $p$-value of 0.05. As shown in Fig. 15 the SVM and MaxEnt with sgd have higher average accuracy than the MaxEnt, so we ran McNemar's test on these two classifiers and the **computed $p$-value was 0.005667**, it indicates that these two classifiers are different, and SVM performs significantly better than the MaxEnt, and the final result is statistically significant.

## Overall results analysis

To examine the performance of our framework and ultimately the classifiers based on our proposed framework, we ran a 5-fold CV, so that in each run 1/5 of the reviews and tweets are held as validation data and remaining held as training data. This setting repeated for every fold and in the last, we checked the F1, precision, and recall scores of our classifiers to check the performance. The detailed measure scores of classifiers while compared with baseline methods are shown in results and analysis sections of the Amazon and social media dataset separately. While analyzing the Amazon dataset results, the appropriate model was with 20 topics, and with bi-gram vectors, it may be because the Amazon dataset contains relatively large texts and more information. When we change the parameters and try the LDA model with bigrams and decrease the topic numbers then it ultimately affects on average F1 score and recall scores of classifiers, as the F1 score drastically decreases from 91% to lowest 64%, similarly recall scores drops to 65% from the 91%. We compared our approach with the baseline approach such as without topic distributions feature vectors, and we implemented the typical text classifiers with different

**Table 8** Comparative results of Best Average F1,Precision and Recall score with baseline approaches.

| Methods | F1 Score | Precision | Recall |
|---|---|---|---|
| TFIDF SVM SGD | 82% | 83% | 80% |
| TFIDF Multinomial NB | 74% | 76% | 75% |
| TFIDF MaxEnt | 71% | 72% | 68% |
| doc2vec MaxEnt | 86% | 77% | **92%** |
| T2F (SVM) | **91%** | 87% | 91% |
| T2F (MaxEnt) | 81% | 83% | 78% |
| T2F (MaxEnt sgd) | 88% | **91%** | 88% |

embedding schemes such TFIDF and doc2vec as a baseline, and overall our approach fairly performed well in classifying the text. In Table 8, baseline method overall highest achieved scores are given in comparison with T2F approach, doc2vec applied on Amazon dataset performs best among the baselines, but still, it is 5% lower than the proposed method in terms of f1 score, T2F, when applied with SVM classifier, achieved the best outcomes and yield 91% f1 score, on MaxEnt sgd classifier it achieved 88% f1 score, still 2% higher than the highest score of implemented baseline method which was 86%. To further analyze from the perspective of different types of embedding schemes researchers proposed in their studies, we have decided to compare our proposed approach with approaches from other research studies that classify the textual data, we have picked up performance results from those research studies, those leveraged different feature embedding schemes, and compared it with our proposed approach, we compared with those approaches from prior studies because their context was also to classify the data by using different feature vectors schemes, like researchers in (*Masood & Abbasi, 2021*) used graph embedding features to classify the social media textual data (284k tweets) into 3 categories and highest overall F1 score was 87% as compared to 91% F1 score of our framework, see results in Table 9, for the dataset, they manually collected the tweets, manually labeled them into categories of rebel users and classify them. Another study uses the feature vector embedding combining initial letter, paragraph, and frequency features to classify the English documents (174 documents) of 4 different categories and their F1 scores fall short by 7% and 5% while using MaxEnt and support vector machine algorithms respectively (*Luo, 2021*) in comparison with our proposed T2F. Graph of words and subgraph feature representations experimented instead of a typical bag of word features by (*Rousseau, Kiagias & Vazirgiannis, 2015*) and maximum F1 score reached up to 79% while implementing on Amazon dataset (16000 user reviews), their dataset is related to our Amazon review dataset to some extent, instead, they just used the portion of Amazon reviews dataset only about specific products categories such as Kitchen, DVD's, books and electronics and we are using Amazon reviews about all the products. All the comparisons with baseline approaches and some other proposed approached in different research studies imply that our novelistic framework performed fairly better. It also indicates that apply the topic modeling with more topics when you have large sentence texts. This can be applied to other classification problems, online complaints, document classification, news classification, and medical text classification.

**Table 9  Comparative results of Best Average F1, Precision and Recall score with prior studies work from the perspective of using different feature representation embedding schemes.**

| Prior study Methods | F1 Score | Precision | Recall |
|---|---|---|---|
| SRI (profile, content+ graph) masood15using | 87% | 91% | 90% |
| SRI (profile, content) masood15using | 79% | 79% | 79% |
| SRI doc embedding masood15using | 86% | 87% | 88% |
| IPF SVM LUO20213401 | 86% | 88% | 87% |
| IPF with MaxEnt LUO20213401 | 81% | 83% | 85% |
| T2G embeddings svm rousseau2015text | 79% | 79% | 77% |
| **Proposed Methods** | | | |
| T2F (SVM) | **91%** | 87% | 91% |
| T2F (MaxEnt) | 81% | 83% | 78% |
| T2F (MaxEnt sgd) | 88% | **91%** | 88% |

# DISCUSSION

A novel framework with the integration of topic distribution features from an unsupervised topic modeling approach considering the features selection is presented. It deals with sparse, user-oriented, short, and slang types of data from different domains. Relevant features extraction to increase the classifier performance is the main purpose of this framework. We focus on the semantic unsupervised generated structure of words that occurred in the texts to classify the user reviews or tweets and how it can assist in supervised classification. Besides classification of user reviews or tweets, recent studies concatenated recently evolved doc2vec with LDA topics, researchers in *Mitroi et al. (2020)* proposed topicdoc2vec model for classifying the sentiment from textual data, they applied doc2vec for vectorizing the textual content and LDA to detect topics, and then they combined both doc2vec vector representation of the best topic of the document through LDA and named it as topicdoc2vec. Their approach claimed to be an approach that adds the context of the topic to the classification process. Although it is an effective approach to construct the context of a document through combined embeddings, this can also be done by only converting LDA topics and their probability distributions to feature vectors as we did in our framework, which is easy to use, flexible, and gives good classification performance. Most of the researchers leverages LDA in combination with other techniques to create a joint topic-oriented word embeddings for a specific context; researchers in *Geetha (2020)* built a joint topical model through LDA, and that model associates topics with a mixture of distributions of words, hashtags and geotags to create topical embeddings specifically for location context. Their embeddings with co-occurrence and location contexts are specified with hashtag vector and geotag context vector respectively. Indeed it is an interesting approach to explore the LDA topic model more for creating embeddings, but it is specified and restricted to geo-located textual data.

Our framework gives comparatively better results in comparison to other prior study approaches that used typical features such as TFIDF, graph embeddings, and graph of words features (see Fig. 16) and with baseline approaches comparison (see Fig. 17). In this framework, we did not feed the classifier with classic TFIDF representations or doc2vec

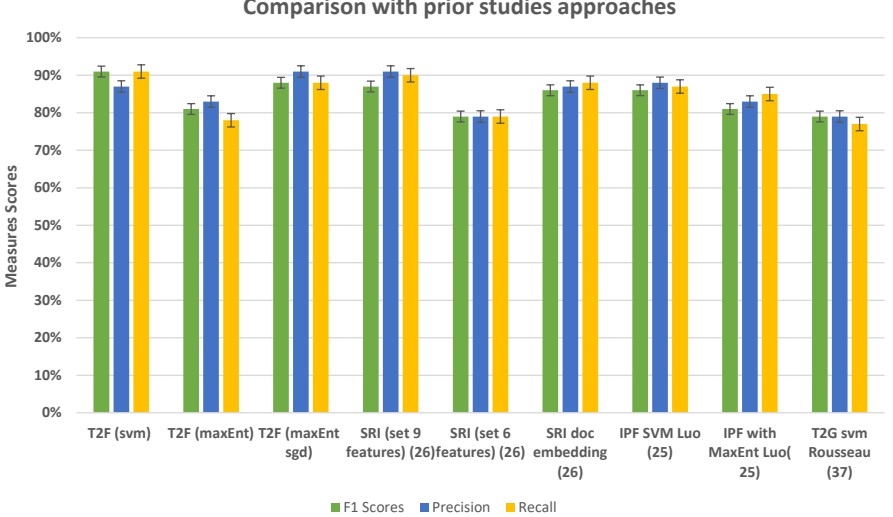

**Figure 16** **Comparative results of evaluation measures in comparison with prior studies approaches.**

word embeddings, instead, we feed classifiers with novel topic distributions features after getting topics on the dataset. This approach can be seen as semi-supervised in a way that it feeds the feature vectors from the unsupervised topic modeling approach into supervised classification algorithms. While building LDA models we analyzed that to extract the most relevant topic distributions, careful text preprocessing is very necessary as it ultimately impacts the model performance. In this regard, we leveraged lemmatization instead of stemming in our text preprocessing, because it gives or reduced the words into their root form with the contextual meaning (*Ullah et al., 2020*). This framework is flexible in a way that it only requires text contents and categories in which you want to classify the data, and this framework is capable to be applied to different domains, like opinion mining, social media sentiment classification, user reviews classification, customer complaints classification government organization. From the results, we found out that for large type text such as documents and large reviews LDA model with more topics would be more suitable and for the sparse short and slang type of texts, an LDA model with fewer topics would be feasible.

## CONCLUSION AND FUTURE WORK

The proposed framework implements an LDA topic model with text classifiers, which can make a text classification by leveraging the hidden topics retrieved from datasets. The method was tested on two datasets of two different domains, datasets with noisy values, sparse data, and imbalanced ratios within, and our proposed method handles that as well in a way to classify the text. From the results, it is evident that our method outperforms other baseline approaches and comparable methods by a reasonably good margin in terms of average F1 scores. We have measured the validity of our model through the 5-fold CV that yields 81% classification accuracy, 5X2 fold CV paired *t*-test, and McNemar's test

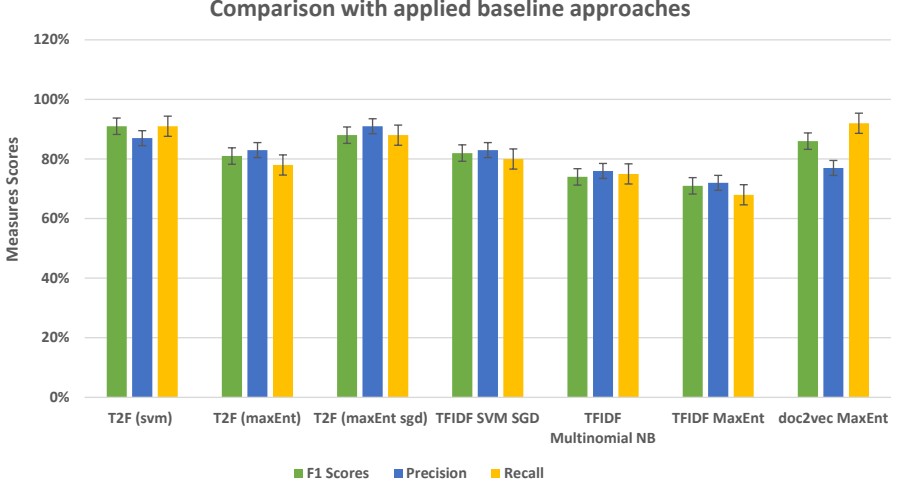

**Figure 17** Comparative results of evaluation measures in comparison with baseline approaches.

statistics. In addition, we applied our model on unseen data, which includes utilizing the topic distributions from specific year's data and applying it to completely unseen data, and this behavior also gives good results in terms of evaluation measures performance. When compared with baseline and prior study approaches, results show improvement while using T2F representations, with the highest 91% average F1 score with SVM classifiers along with bi-grams, and the highest mean accuracy of 81%. Moreover, the search for the best combination of parameters is based on how evaluation measures are performed. We got the best combination of SVM classifiers using bi-grams on Amazon dataset that yields highest average F1 score, and with MaxEnt classifier with both 15 and 10 topics and trigrams combination that gives highest average F1 score on social media dataset, and then on 5-fold CV evaluation the MaxEnt sgd classifier with bigrams and 20 topics gives best mean accuracy results. We find that our T2F model outperforms other baselines and prior study approaches on average F1 score and mean accuracy; overall, our framework performs better if we see evaluation measures results, which indicate that topic-oriented features can be leveraged as one of the features representation techniques while classifying the texts. Also with these findings, we prepared a model that paved the way to create topic-oriented features (T2F) representation of content for classification; it can be applied into any text classification context. Furthermore, we have demonstrated that text representation based on LDA topic modeling has more semantic meaning and can improve the classification performance while performing in a semi-supervised manner. Many improvements can be made, such as one can apply this method on medical domain datasets. In the future, we will extend our framework to automatic labeling of data to prepare a labeled dataset to be used in supervised algorithms. We will gather the topic distributions and apply ranking algorithms and analyze the topics in terms of weightage and label the documents, reviews or tweets; this will reduce the cost of human labels and will also remove the need of gathering the labeled datasets, because not every public dataset has labels. Also, while

applying classification on the labeled dataset, we will explore some deep learning classifiers such as used by *Olteanu et al. (2014)* and will investigate the impact of these classifiers on classification performance.

### Funding

This work was supported by the National Key Technologies R&D Program (under grant number 2020YFB1712401, 2018YFB1701401), the Nature Science Foundation of China (grant number 62006210), the major project of Zhengzhou Collaborative Innovation (under grant number 20XTZX-009, 20XTZX-X010), the National Key R&D Program of China 2018 and the Key Scientific and Technological Research Projects in the Henan Province of China under grant number 192102310216, the National Key R&D Program of China (2018\*\*\*\*\*02), and the 2020 Major Project Public Benefit Project in Henan Province (201300210500). The funders had no role in study design, data collection and analysis, decision to publish, or preparation of the manuscript.

### Grant Disclosures

The following grant information was disclosed by the authors:
The National Key Technologies R&D Program: 2020YFB1712401, 2018YFB1701401.
The Nature Science Foundation of China: 62006210.
The major project of Zhengzhou Collaborative Innovation: 20XTZX-009, 20XTZX-X010.
The National Key R&D Program of China 2018.
The Key Scientific and Technological Research Projects in the Henan Province of China: 192102310216.
The National Key R&D program of china: 2018\*\*\*\*\*02.
The Major Public Benefit Project in Henan Province: 201300210050.

### Competing Interests

The authors declare there are no competing interests.

### Author Contributions

- Junaid Abdul Wahid conceived and designed the experiments, performed the experiments, analyzed the data, performed the computation work, prepared figures and/or tables, authored or reviewed drafts of the paper, and approved the final draft.
- Lei Shi analyzed the data, prepared figures and/or tables, authored or reviewed drafts of the paper, and approved the final draft.
- Yufei Gao conceived and designed the experiments, performed the experiments, analyzed the data, performed the computation work, prepared figures and/or tables, authored or reviewed drafts of the paper, and approved the final draft.
- Bei Yang performed the experiments, authored or reviewed drafts of the paper, and approved the final draft.
- Yongcai Tao analyzed the data, authored or reviewed drafts of the paper, and approved the final draft.

- Lin Wei analyzed the data, authored or reviewed drafts of the paper, and approved the final draft.
- Shabir Hussain performed the experiments, prepared figures and/or tables, and approved the final draft.

## Data Availability

Source codes are available in the Supplemental Files.

We used public datasets:

- Amazon dataset: https://www.kaggle.com/snap/Amazon-fine-food-reviews.
- Social media dataset 6 disasters: https://github.com/sajao/CrisisLex/tree/master/data/CrisisLexT6/.
- Social media remaining one disaster dataset (Resource #2): https://crisisnlp.qcri.org/.

## Supplemental Information

Supplemental information for this article can be found online at http://dx.doi.org/10.7717/peerj-cs.677#supplemental-information.

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
