# Peer review of "Topic2features: a novel framework to classify noisy and sparse textual data using LDA topic distributions"

_PeerJ Computer Science, doi:10.7717/peerj-cs.677_

## Round 0.1 · original submission · Major Revisions

Please revise your manuscript addressing all the comments and suggestions by reviewers.

Reviewer 1 ·

Basic reporting

The language of the manuscript is clear and unambiguous. The paper is structured well, including figures, tables and shared raw data.

Experimental design

The research questions are well defined, meaningful and relevant.

Validity of the findings

The experimental results are well stated.

Additional comments

In this paper, a novel classification scheme based on LDA topic modelling has been presented to classify noisy and sparse textual data. In general, the paper is structured well and contributes to the literature. However, a number of issues listed below should be taken into consideration to improve the content of the paper:
1- The language of the manuscript should be enhanced.
2- The manuscript lacks of discussing recent studies (2020-2021 papers) on LDA2vec.
3- The empirical results should include statistical validity tests.

Reviewer 2 ·

Basic reporting

The authors present a text classification scheme enriched with LDA-based features.
Overall, the manuscript was ambiguous at some sections (see attached PDF), however, the overall idea is clear.

The references are clearly provided, however, please re-check that the citation style is consistent (sometimes you have just numbers, e.g., (14)).

Article structure is OK, with the exception of some low-res images.

Experimental design

The authors evaluate their method first by exploring different hyperparameter settings that offer sufficient performance. In the second step, they compare to existing baselines. The first part is done adequately, however, I have concerns related to the second part of the evaluation.

Here, the authors compare their method to "baselines", however it remains unclear whether they actually run the experiments against the baseline methods or just picked up performances from the literature. In the latter case, please additionally clarify why the results are comparable. Furthermore, I would suggest using at least one of the following baselines other researchers are familliar with, so it is clear that your method actually works:
1.) doc2vec + LR (scikit-learn)
2.) BERT (end-to-end)
3.) BERT (sentence-bert) + LR (scikit-learn)

Comparison against these could shed additional light on whether the proposed method performs well (and when). Finally, the number of data sets could be larger, you are only exploring the reviews, however I would be willing to believe there is potential if the proper comparisons are conducted.

Validity of the findings

The findings are in alignment with the empirical evaluation as it stands -> the proposed method is superior to others. One problem I have with the claim is statistical significance claim: You claim that the results are significantly bettter. I saw no statistical tests being conducted to verify/prove this claim.

Conclusions could be longer (see attached PDF).

Additional comments

I've left numerous comments also in the attached PDF, which will hopefully improve the manuscript's quality.

Annotated reviews are not available for download in order to protect the identity of reviewers who chose to remain anonymous.

Reviewer 3 ·

Basic reporting

The article is a proposal on framework called Topic2features. Generally, it has been well-written in terms of paper organization and writing style. The framework has been described in detail, has been tested with amazon dataset, and yet has been compared and evaluated with few algorithms and benchmark.

I have no major issues to accep the paper as it is.

Experimental design

accceptable

Validity of the findings

acceptable

Additional comments

Very good paper, and well written.

Reviewer 4 ·

Basic reporting

The paper proposed a novel framework topic2features (T2F) to deal with short and
sparse data using the topic distributions of hidden topics gathered from the dataset and converting them into features to build a supervised classifier.

It seems that the merit of the study is very interesting and it contains a novel of a new approach of representation that is based on topics instead of a bag of words.
The authors fail to give a clear idea of the main approach of this study. The approach is described a very vague and it is hard to get how the topics were used in a vector space.
The study needs to be improved dramatically with more description of the algorithm using pseudocode or more clear workflows and figures.
I will not be able to judge the content of the study without a clear understanding of the novelty of topic representations. This section should be improved.
Also, I would suggest to the author to send the article for English proof before resubmitting it.

Experimental design

The study shows satisfactory experiments

Validity of the findings

I will not be able to judge the content of the study without a clear understanding of the novelty of topic representations. This section should be improved.

Additional comments

Line 287 : might me should be "might be"
Figure 6 should be replaced with a more clear image or just replace with a regular table.
Figure 7 should be replaced with a more clear image or just replace with a regular table.

---

## Round 0.2 · Minor Revisions

The manuscript requires some minor revisions. Authors are required to address the suggested modification in the manuscript, especially improvements in the English language.

Reviewer 4 ·

Basic reporting

The manuscript in its second round is clear and unambiguous. I have a concern about English as it seems that the English is very simple and needs to be improved to a higher level.
The structure of the article is easy to follow, while it includes many figures that help to understand the idea of the algorithm.

Experimental design

The research question is wee defined and relevant. The article explains nicely the solution to the given question.

Validity of the findings

The article presents performance and validation approaches that were applied to the suggested algorithm.
The results are statistically sound.

Additional comments

line 375: and -> should be just and
line 261 I would use a number instead of no
The authors supposed to use mathematical styling in writing the mathematical phrases and equations. As such KI in line 299, line 310

---

## Round 0.3 · accepted · Accept

Manuscript is accepted for the publication.